# Peroxisome Proliferator-Activated Receptor Gamma (PPARγ) Suppresses Inflammation and Bacterial Clearance during Influenza-Bacterial Super-Infection

**DOI:** 10.3390/v11060505

**Published:** 2019-06-01

**Authors:** Radha Gopal, Angelico Mendy, Michael A. Marinelli, Lacee J. Richwalls, Philip J. Seger, Shivani Patel, Kevin J. McHugh, Helen E. Rich, Jennifer A. Grousd, Erick Forno, John F. Alcorn

**Affiliations:** 1Department of Pediatrics, UPMC Children’s Hospital of Pittsburgh, Pittsburgh, PA 15224, USA; radha.gopal4@chp.edu (R.G.); mam637@pitt.edu (M.A.M.); ljr40@pitt.edu (L.J.R.); pjs75@pitt.edu (P.J.S.); shivaninickpatel@gmail.com (S.P.); mchughkj@upmc.edu (K.J.M.); her39@pitt.edu (H.E.R.); jag282@pitt.edu (J.A.G.); erick.forno@chp.edu (E.F.); 2College of Public Health, University of Iowa, Iowa City, IA 52246, USA; angelico-mendy@uiowa.edu

**Keywords:** lung, pneumonia, MRSA, PPARγ agonist, rosiglitazone, neutrophils, cytokines, chemokines

## Abstract

Influenza virus is among the most common causes of respiratory illness worldwide and can be complicated by secondary bacterial pneumonia, a frequent cause of mortality. When influenza virus infects the lung, the innate immune response is activated, and interferons and inflammatory mediators are released. This “cytokine storm” is thought to play a role in influenza-induced lung pathogenesis. Peroxisome proliferator-activated receptor gamma (PPARγ) is a member of the nuclear hormone receptor super-family. PPARγ has numerous functions including enhancing lipid and glucose metabolism and cellular differentiation and suppressing inflammation. Synthetic PPARγ agonists (thiazolidinediones or glitazones) have been used clinically in the treatment of type II diabetes. Using data from the National Health and Nutrition Examination Survey (NHANES), diabetic participants taking rosiglitazone had an increased risk of mortality from influenza/pneumonia compared to those not taking the drug. We examined the effect of rosiglitazone treatment during influenza and secondary bacterial (Methicillin resistant *Staphylococcus aureus*) pneumonia in mice. We found decreased influenza viral burden, decreased numbers of neutrophils and macrophages in bronchoalveolar lavage, and decreased production of cytokines and chemokines in influenza infected, rosiglitazone-treated mice when compared to controls. However, rosiglitazone treatment compromised bacterial clearance during influenza-bacterial super-infection. Both human and mouse data suggest that rosiglitazone treatment worsens the outcome of influenza-associated pneumonia.

## 1. Introduction

Influenza, while being a common illness, takes a heavy toll on the healthcare system as the Center for Disease Control and Prevention estimates that between 12,000 and 56,000 deaths have occurred annually in the United States since 2010. The 2017–2018 influenza season was one of the worst influenza seasons in recent history in terms of mortality, which was approximated at 80,000 in the United States. In addition to seasonal influenza outbreaks, there is global pandemic potential, including four recorded instances over the past century, including 2009. 

Influenza induces Type I, Type II, and Type III interferons, which are responsible for interfering with viral replication in infected cells [1,2,3,4,5]. Binding of Type I interferons to their transmembrane protein receptors results in the formation of STAT1–STAT2 heterodimers, while binding of Type II interferons signal through STAT1–STAT1 homodimers, translocate into the nucleus, and sub-sequentially bind to the promoter element of interferon stimulated genes (ISG) to control the virus [6,7]. Mx proteins are interferon-induced GTP binding proteins that are part of the type I and type III IFN-induced antiviral response [8]. Mx1, inhibits influenza viral transcription and replication by suppressing the polymerase activity of viral ribonucleoproteins [9]. Influenza infection induces a variety of inflammatory cytokines and chemokines such as IL-6, IL-8, TNFα, CCL2, CCL5, CXCL10, and recruitment of neutrophils and macrophages to clear the virus. Neutrophil elastase (ELANE) and cathepsin G (CTSG) are proteolytic enzymes present in neutrophilic azurophilic granules that are involved in antimicrobial activity [10,11]. However, excessive inflammatory immune responses from neutrophils, macrophages and proinflammatory cytokines drive lung pathology [1,12]. Although most influenza infections result in mild to moderate disease, secondary bacterial infection results in high rates of mortality and morbidity. Community-acquired methicillin-resistant *Staphylococcus aureus* (MRSA) has been shown as a significant source of mortality during influenza-associated secondary bacterial infection [13,14,15]. Influenza infection predisposes the lung to secondary bacterial infection by dysregulation of innate and adaptive immune responses [16,17,18,19,20,21,22]. We have shown that type I IFN and STAT1 inhibit the *S. aureus*-induced Type 17 response, thereby decreasing bacterial clearance during influenza-bacterial super-infection [23,24,25]. Also, our recent study has shown that the STAT2 signaling suppress macrophage activation and bacterial clearance during influenza-bacterial super-infection [26]. Given the mortality associated with influenza and influenza-bacterial super-infection, research has focused on identification of therapeutic targets.

PPARγ is a member of the nuclear hormone receptor super-family and acts in the nucleus as a transcription factor. PPARγ is known to enhance lipid and glucose metabolism as well as cellular differentiation. Additionally, PPAR-γ has been shown to inhibit expression of inflammatory cytokines and produce an anti-inflammatory response in multiple models of disease [27,28,29]. PPARγ is activated by natural ligands such as 15-deoxy-D12; 14-prostaglandin J2 (15d-PGJ2) or synthetic ligands including thiazolidinediones (TZDs) or glitazones, and GW1929 [27,30,31]. Thiazolidinediones (rosiglitazone or pioglitazone) have been used clinically in the treatment of type II diabetes mellitus. These glitazones can be used as a monotherapy or in combination with insulin, metformin or sulfonylureas [32,33,34]. Also, studies have shown that rosiglitazone increases the anti-inflammatory immune response during acute pulmonary inflammation [35,36]. Recently we have shown that metformin is associated with a decrease in chronic lower respiratory disease (CLRD)-related mortality [37]. Since large numbers of patients have been treated with rosiglitazone, we used data from the National Health and Nutrition Examination Surveys (NHANES) conducted between 1988–1994 and 1999–2010 and determined the influenza/pneumonia mortality associated with the medication. We then hypothesized that increasing PPARγ activation will decrease the inflammatory immune response during influenza and influenza-bacterial super-infection. In this study, we examined the role of rosiglitazone treatment during influenza infection and influenza-bacterial super-infection by determining influenza viral burden, viral-induced interferons and their downstream stimulated genes, inflammatory cellular responses, and pro- and anti-inflammatory cytokines in the lung.

## 2. Materials and Methods

### 2.1. Animals

Mice (C57BL/6) were purchased from Taconic Farms. All the experimental mice were sex matched (male) and used between 6–8 weeks of age. All the animal experiments were performed according to the University of Pittsburgh Institutional Animal Care and Use Committee guidelines (IACUC), protocol 17071194, dated, 06/01/2018.

### 2.2. Experimental Infections and Animal Treatments

All infections were performed on isoflurane-anesthetized mice via oropharyngeal aspiration. Mice were infected with 100 plaque-forming units (PFU) of influenza A/PR/8/34 (influenza H1N1) in 50 µl of sterile PBS. Methicillin-resistant *S. aureus* (USA300) was cultured in casein hydrolysate yeast extract-containing modified medium for 18 h to stationary growth phase at 37 °C and diluted to an infectious inoculum of 5 × 10^7^ CFU in 50 µL of sterile PBS. For super-infection experiments, mice were first challenged with influenza (100 pfu) or vehicle and then infected with MRSA or vehicle on day 6 after influenza infection [25,26]. In some experiments, mice were treated with rosiglitazone (10 mg/mL) or vehicle (DMSO) from day 0–6 of influenza infection via intraperitoneal injection. 

### 2.3. Measurement of Lung Inflammation

Mice were harvested 7 days after influenza infection. In time course experiments, mice were harvested every other day up to 14 days. In super-infection experiments, mice were harvested twenty-four hours following bacterial challenge, 7 days post-influenza. Bronchoalveolar lavage (BAL) fluid was collected with 1 mL of sterile PBS and cell differential counts were performed [24,26,38]. The cranial lobe of the right lung was homogenized in PBS and used to determine the bacterial colony and cytokine analysis. The middle and caudal lobes of the right lung were used for RNA isolation using the Absolutely RNA Miniprep Kit (Agilent Technologies, Santa Clara, CA, USA). Gene expression was analyzed by RT-PCR utilizing commercially available Taqman primer and probe sets (Applied Biosystems, Foster City, CA, USA). Fold changes in mRNA expression were calculated using the delta-delta CT method, and were normalized to the endogenous housekeeping gene hypoxanthine-guanine phosphoribosyl transferase (HPRT). 

### 2.4. Flow Cytometry

Whole mouse lungs were digested in collagenase and passed through 70 μm filters as described [26]. Single cell suspensions were stained with CD11b (M1/70), CD11c (HL3), Ly6C, Ly6G. Cells were collected in a Becton Dickinson FACS Aria flow cytometer with FACS Diva software (BD, Franklin Lakes, NJ, USA). Flow cytometric analysis was performed using FlowJo (Tree Star, Ashland, OR, USA).

### 2.5. Statistical Analysis

The data analyses were performed using GraphPad Prism Software. All the data were presented as mean ± SEM. Differences between two groups were analyzed using two-tailed Student’s *t*-test and multiple experimental groups were analyzed using one-way ANOVA with Tukey’s post-hoc test. Differences were considered significant when *p* ≤ 0.05.

### 2.6. Human Data Source

We used data from the National Health and Nutrition Examination Surveys (NHANES) conducted between 1988–1994 and 1999–2010 by the National Center for Health Statistics (NCHS) of the Centers for Disease Control and Prevention (CDC). The NHANES is a continuous cross-sectional survey that uses a complex multistage sampling design to derive a sample representative of the U.S. population. During this period, 6606 participants with diabetes (defined as taking medication for diabetes or a hemoglobin A1C ≥ 6.5%) provided data on the use of thiazolidinediones and other antidiabetic drugs and were followed for mortality through to December 31, 2015. We merged the NHANES files with the corresponding Mortality-Linked Files created by the NCHS using National Death Index records and death certificates. The NHANES protocols were approved by the Institutional Review Boards of the NCHS and the CDC, and informed consent was obtained from each participant. Please see https://www.cdc.gov/nchs/nhanes/irba98.htm for details.

### 2.7. Definition of Variables

NHANES participants were asked about medication use in the past month. Those who reported taking prescription medications were further asked about the name and duration of the product used. The product name was recorded preferentially from the medication container label and if the container was unavailable, it was reported by the participant. The medications we included were rosiglitazone, other thiazolidinediones, insulin, and other oral antidiabetic drugs (metformin, sulfonylureas, other antidiabetics). Influenza/pneumonia mortality was defined using the Tenth Revision of the International Classification of Diseases (ICD-10) (corresponding to the codes J9-J18). Detailed information on the classifications used by the NHANES is available at https://www.cdc.gov/nchs/data/datalinkage/public-use-2015-linked-mortality-files-data-dictionary.pdf. Using a questionnaire, the NHANES also collected data on covariates including age, gender, race/ethnicity, family poverty income ratio (PIR), cigarette smoking, and previous diagnosis of asthma or COPD.

### 2.8. Human Data Statistical Analysis

Descriptive analyses were performed by rosiglitazone status. Cox proportional hazards regression was used to estimate the hazard ratio (HR) and 95% confidence interval (CI) for the risk of influenza/pneumonia mortality associated with the medication. The model was adjusted for age, gender, race/ethnicity, PIR, cigarette smoking, asthma or COPD, treatment by insulin, and treatment by other oral antidiabetic drugs. The analyses were performed in SAS (Version 9.4) accounting for NHANES sampling weights and complex design and *P* values <0.05 were considered statistically significant.

## 3. Results 

### 3.1. Influenza/Pneumonia Mortality Is Increased in Diabetic Patients in Response to Rosiglitazone Treatment

We first determined whether rosiglitazone treatment is associated with mortality from influenza-related pneumonia in humans, using data on diabetic patients from NHANES. Among the 6606 participants included in the analysis, the prevalence of rosiglitazone treatment at baseline was 4.6% (adjusted for sampling weight and study design). During a median follow-up of 9.2 years, 68 participants died of influenza/pneumonia. Diabetic subjects taking rosiglitazone and those not taking the drug did not differ by age groups, gender, race/ethnicity, PIR, cigarette smoking, or asthma/COPD prevalence. However, they had a higher prevalence of treatment with insulin and other oral antidiabetic drugs (Table 1). The mortality rate from influenza/pneumonia during follow-up was more than twice as high in participants taking rosiglitazone (2.0 per 1000 person-years) than in those not taking the drug (0.7 per 1000 person-years). Adjusted Cox regression showed that rosiglitazone was associated with a ~4-fold increased risk of mortality from influenza/pneumonia (HR: 4.09, 95% CI: 1.31–12.82, *P* = 0.016) (Figure 1). We found no association between treatment with other oral antidiabetic drugs (metformin, sulfonylureas, or others) and the outcome. These data suggest that rosiglitazone may be detrimental in influenza/pneumonia.

### 3.2. PPARγ Expression Is Suppressed in Response to Influenza Infection

PPARγ is known to have an anti-inflammatory role in several disease models [29,39,40,41,42]. However, the role of PPARγ during influenza infection is not clear. To determine the relationship between PPARγ expression and influenza infection, we infected mice with influenza and measured the mRNA expression of PPARγ and influenza viral burden at days 4, 8, and 12 post-infection by RT-PCR. We found the viral titer was highest on day 4 when compared to day 8 and 12; virus was not detectable on day 12 after influenza infection (Figure 2B). However, PPARγ expression is decreased throughout influenza infection progression (day 4 until day 12) (Figure 2A). These data demonstrate that PPARγ expression is decreased during influenza infection and suggest that exogenous PPARγ agonists may have an impact on inflammation. 

### 3.3. Rosiglitazone Treatment Suppresses Influenza Viral Burden and Reduces Inflammatory Cells in BAL during Influenza Infection

Several studies have shown that PPARγ agonists decrease the inflammatory response during acute pulmonary infection [35,36,43]. We determined whether boosting PPARγ activity has an anti-inflammatory effect during influenza infection. We infected mice with influenza and administered rosiglitazone or vehicle (DMSO) once per day from 0–6 days post-infection. We found that influenza burden (influenza M protein expression) significantly decreased in response to rosiglitazone treatment (Figure 3A). Further, the expression of interferons (IFNβ and IFNγ), transcription factors (STAT1, STAT2) and interferon stimulated genes (ISG) also significantly decreased in mice treated with rosiglitazone when compared to vehicle treatment (Figure 3B–F). However, we found no differences in weight loss between the vehicle or rosiglitazone treatments during influenza infection (Appendix A). These data suggest that PPARγ agonists may limit influenza viral load and interferon responses.

Next, to examine the inflammatory response, we stained cells from bronchoalveolar lavage (BAL) and performed differential counting. We found decreased neutrophil and macrophage cells in the BAL of rosiglitazone-treated mice when compared to the vehicle-treated controls (Figure 4A,B). Further, the number of lymphocytes also trended to decrease in rosiglitazone-treated mice when compared to controls (Figure 4C). Next, we determined the effect of rosiglitazone treatment on neutrophil activity by measuring the gene expression of cytochrome B (CYBB), cathepsin G (CTSG and neutrophil elastase (ELANE). We found decreased expression of these genes in response to rosiglitazone treatment (Figure 4D–F). These findings are consistent with PPARγ agonist impairment of inflammation.

We then determined the effect of rosiglitazone treatment on induction of pro-and anti-inflammatory cytokines and chemokines. We found decreased gene expression of IL-6, IL-12p40, CCL2, CXCL9 and CXCL10 in the rosiglitazone-treated group when compared to the vehicle-treated group (Figure 5A–E). Further, we found trends towards decreased protein levels of IL-6, IL-12p40, IL-12p70, CCL2, CCL3, and CCL4 in the rosiglitazone treatment group (Figure 5F). However, there was no difference in potentially anti-inflammatory cytokines IL-4, IL-10, and IL-13 (Figure 5G). Together, these data suggest that rosiglitazone treatment decreases influenza viral burden, antiviral immune signaling, and the inflammatory cellular response during influenza infection. 

### 3.4. PPARγ Agonist Treatment Compromises Bacterial Clearance during Influenza-Bacterial Super-Infection

The suppression of pro-inflammatory responses induced by PPARγ agonist during influenza infection is not consistent with increased rosiglitazone associated mortality seen in humans. Influenza mortality in humans is often associated with secondary bacterial pneumonia. Next, we addressed the effect of rosiglitazone treatment during influenza-bacterial super-infection pneumonia. We infected mice with influenza and administered rosiglitazone or vehicle (DMSO) once per day from 0–6 days post-infection. On day 6 we infected mice with MRSA and harvested tissues one day later. Interestingly, we found that super-infected, rosiglitazone-treated mice harbored more lung bacteria when compared to super-infected, vehicle-treated mice (Figure 6A). Similar to influenza infection, there were no differences in weight loss observed between vehicle and rosiglitazone treatment during influenza-bacterial super-infection (Appendix A). Next, we determined whether rosiglitazone treatment decreases influenza viral burden during super-infection. We found decreased expression of influenza M protein in the lungs of rosiglitazone-treated super-infected mice when compared to vehicle-treated super-infected mice (Figure 6B). However, we found no significant differences in the gene expression levels of IFNβ, IFNγ, STAT1, and STAT2 in rosiglitazone-treated mice when compared to controls (Figure 6C–F). Further, we found decreased expression of the ISG, Mx1 in rosiglitazone treatment group when compared to controls (Figure 6G). These data suggest that PPARγ agonist inhibition of interferon responses is muted during bacterial super-infection.

Next, we determined the inflammatory cellular response in the airspace by measuring the number of neutrophils, macrophages and lymphocytes in BAL. We found decreased numbers of neutrophils, macrophages and lymphocytes in super-infected, rosiglitazone-treated mice when compared to super-infected, vehicle-treated mice (Figure 7A–C). Further, we determined whether neutrophil activity is decreased in response to rosiglitazone treatment. We found decreased expression of CTSG and ELANE in super-infected, rosiglitazone-treated mice when compared to super-infected, vehicle-treated mice (Figure 7D–F). We also found a decreased percentage of neutrophils in the lung by flow cytometry (Figure 7G,H). These data suggest that rosiglitazone treatment decreases the cellular response to bacterial super-infection by reducing neutrophil numbers and activity during influenza-bacterial super-infection. 

Finally, we determined whether pro-inflammatory cytokine and chemokines are suppressed in response to rosiglitazone treatment. We found no differences in the gene expression levels of IL-6, IL-12p40, CCL2, CXCL9, and CXCL10 in both rosiglitazone treatment and vehicle controls (Figure 8A–E). We also found that protein levels of IL-6, IL-12p40, IL-12p70, CCL2, CCL3 and CCL4 were not suppressed in response to rosiglitazone treatment during super-infection (Figure 8F). Further, we found the levels of IL-4, IL-10 and IL-13 were not elevated in response to rosiglitazone treatment during super-infection (Figure 8G). These results imply that rosiglitazone treatment specifically acts on neutrophils to reduce their number and activity during influenza-bacterial super-infection, but does not consistently inhibit inflammation, unlike influenza single infection.

## 4. Discussion

In response to influenza viral entry, the host immune system recognizes and induces interferons, recruits immune cells, and induces pro-inflammatory cytokines and chemokines to clear the virus. However, excessive inflammatory cellular recruitment along with cytokine and chemokine production results in lung pathology, morbidity and mortality. Several studies have shown that immunomodulatory agents can decrease the lung inflammatory response without affecting the viral clearance [44,45]. In this study, we found that influenza infection suppressed PPARγ expression on days 4, 8, and 12 post-infection. Despite viral burden not being detectable on day 12, influenza-induced inflammatory cytokines and chemokines are a possible mechanism involved in suppression of PPARγ late during influenza infection. It has been shown that LPS treatment suppress PPARγ expression in macrophages [46]. Also, studies have shown that IFNγ regulates PPARγ expression in adipocytes and macrophages [46,47]. Further, we found that IFNγ suppresses PPARγ expression during influenza and influenza-MRSA super-infection in mice [48]. These data suggest that inflammatory mediators, possibly IFNγ, may be the mechanism involved in suppression of PPARγ following viral clearance.

Next, we sought to determine the effect of rosiglitazone treatment on viral clearance, cellular responses, and inflammatory cytokine and chemokine responses during influenza and influenza-bacterial super-infection. Our results demonstrate that rosiglitazone treatment decreased viral burden, the number of neutrophils and macrophages, and the levels of proinflammatory cytokines and chemokines during influenza infection. The decrease in viral burden in response to rosiglitazone treatment is in accordance with previous finding by Cloutier et al. The authors have shown that the natural PPARγ agonist (15d-PGJ2) decreased viral titers during influenza infection [49]. The reduction in influenza burden due to the PPARγ agonist treatment might be due to the restriction of viral entry in epithelial cells or could be due to the enhanced antiviral function of natural killer or cytotoxic T cells. We recently found a lack of or a small fold viral titer change in STAT1 or STAT2 gene deficient mice infected with influenza infection. [24,26]. These studies suggest redundant antiviral and/or clearance mechanisms during influenza infection. Next, our results showed that rosiglitazone treatment decreased the expression of IFNγ during influenza infection. This is in accordance with the findings of Cunard et al. which showed PPARγ ligands repress the IFNγ promoter by interfering with c-Jun activation [50]. We did not observe a significant decrease in IFNβ in our study unlike the previous finding that PPARγ agonists negatively regulate IFNβ production by preventing interferon regulatory factor 3 binding to the IFNβ promoter [51]. Our study further indicated that rosiglitazone treatment decreased downstream signaling agents STAT1, STAT2, and the ISGs Mx1, CXCL9 and CXCL10 during influenza infection. The rosiglitazone effect on interferon signaling was absent or less pronounced during super-infection, which may explain why treated mice remained increasingly susceptible to secondary bacterial infection.

Several previous studies have found that natural and synthetic PPARγ agonists suppress the major inflammatory signaling pathways, such as AP-1, STAT and NF-KB signaling and inhibit inflammatory cytokine production [52,53,54,55,56]. Accordingly, our study has shown that rosiglitazone treatment decreased the levels of IL-6, IL-12, CCL-2 and CCL6 during influenza infection. Also, Cloutier et al., have shown that the natural PPARγ agonist (15d-PGJ2) decreased viral titers and decreased expression of IL-6, TNFα, CCL2, CCL3, and CXCL-10 during influenza infection [49]. It has been shown that PPARγ agonists decrease the levels of KC and G-CSF in the lungs in response to aerosolized lipopolysaccharide in mice [35]. Our results demonstrate that rosiglitazone does indeed have anti-inflammatory effects in a preclinical model of influenza infection. This raises the question as to why rosiglitazone is associated with increased influenza/pneumonia mortality.

Severe influenza pneumonia is often associated with secondary bacterial infections. Interestingly, a previous study has shown that PPARγ agonist treatment improves bacterial clearance in *Staphylococcus aureus* skin infection [57]. In the lung, it is possible that PPARγ agonist mediated suppression of neutrophil activation may impair anti-bacterial host defense. Indeed, we observed decreased MRSA bacterial clearance in influenza-bacterial super-infected rosiglitazone-treated mice when compared to vehicle-treated super-infected mice. Moreover, we found no differences in the gene expression levels of IFNβ, IFNγ, STAT1, STAT2, IL-6, IL-12p40, CCL2, CXCL9 and CXCL10, and the protein levels of IL-6, IL-12p40, IL-12p70, CCL2, CCL3 and CCL4 in response to rosiglitazone treatment during super-infection. These data suggest that rosiglitazone may worsen influenza pneumonia outcome by inhibiting bacterial host defense. In this setting rosiglitazone was less effective at reducing inflammatory cytokine production, perhaps mitigating its potential protective anti-inflammatory effect. Also, we found decreased expression of IFNγ, CXCL9 and CXCL10 in mice infected with MRSA and treated with rosiglitazone [58]. This finding in the pneumonia setting differs from *S. aureus* skin infection where PPARγ inhibitors worsened bacterial burden, implying that a PPARγ agonist may be useful [59]. Another previous study has shown that pioglitazone treatment decreased the inflammatory response in influenza-induced exacerbation of chronic obstructive pulmonary disease (COPD) [60]. It is unclear why PPARγ agonist treatment may have different effects in the context of influenza-bacterial super-infections.

In our study, we found that rosiglitazone treatment decreased the percentage of neutrophils and expression of genes related to neutrophil activity in both single and super-infection. Our recent finding that that IL-33 increases neutrophil recruitment and thereby increases bacterial clearance during super-infection implicates mature neutrophil function in bacterial clearance in this context [61]. Also, other studies have shown that rosiglitazone decreased the neutrophilic response to cigarette smoke-induced exacerbation by bacteria infection [62]. Accordingly, our data suggest that the decreased neutrophilic response might be a possible mechanism why we found decreased bacterial clearance during super-infection. Further, the human data from NHANES also show an increase in mortality-associated with rosiglitazone treatment in influenza-associated pneumonia. It is interesting that other anti-diabetic drugs did not have a similar association. It is possible that the effect of rosiglitazone on influenza infection is dependent on insulin usage and diabetes comorbidity. Our pre-clinical animal model did not incorporate these variables, but nonetheless observed an impact of rosiglitazone on bacterial super-infection. Both human and mouse data suggest that even though rosiglitazone decreased the viral burden and inflammation during primary influenza infection, it worsened bacterial clearance during influenza-associated pneumonia. This finding may explain the increased influenza pneumonia mortality observed in diabetic patients taking rosiglitazone versus other drugs.

## Figures and Tables

**Figure 1 viruses-11-00505-f001:**
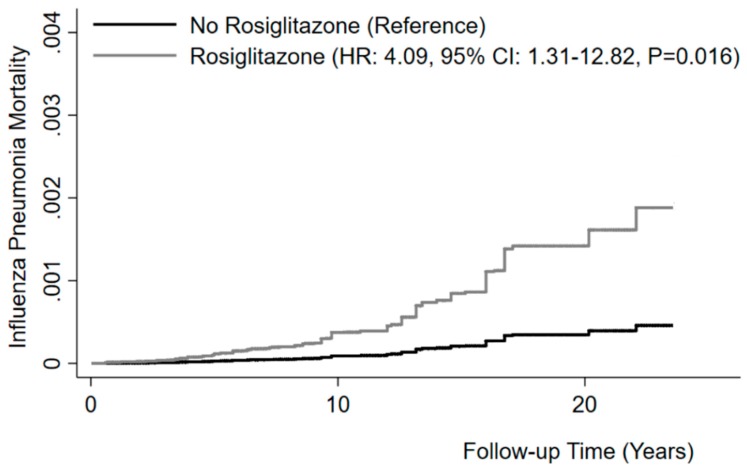
Rosiglitazone and risk of mortality from influenza/pneumonia. Kaplan–Meier curves for cumulative mortality from influenza/pneumonia in non-users and users of rosiglitazone. Model adjusted for age, gender, race/ethnicity, PIR, cigarette smoking, asthma or COPD, treatment by insulin, and treatment by other oral antidiabetic drugs. Model also accounting for competing risk of mortality from causes other than influenza and pneumonia.

**Figure 2 viruses-11-00505-f002:**
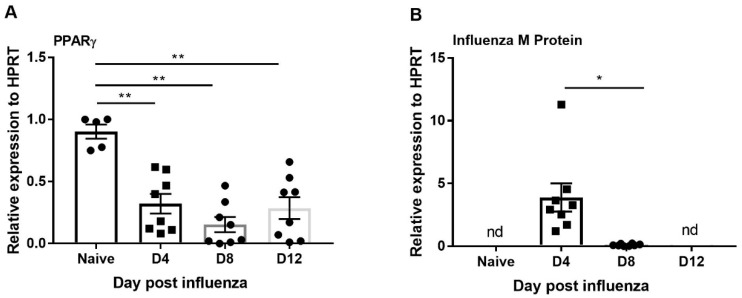
PPARγ expression is suppressed during influenza infection. WT mice were infected with influenza (100 pfu) and harvested from 0, 4, 8, and 12 days post-infection, and (**A**) PPARγ expression and **B**) influenza M protein expression was measured by RT-PCR. *N* = 5–8 per group. Data are represented as mean ±SEM, two tailed Student’s *t* test, * *p* < 0.05, ** *p* < 0.001.

**Figure 3 viruses-11-00505-f003:**
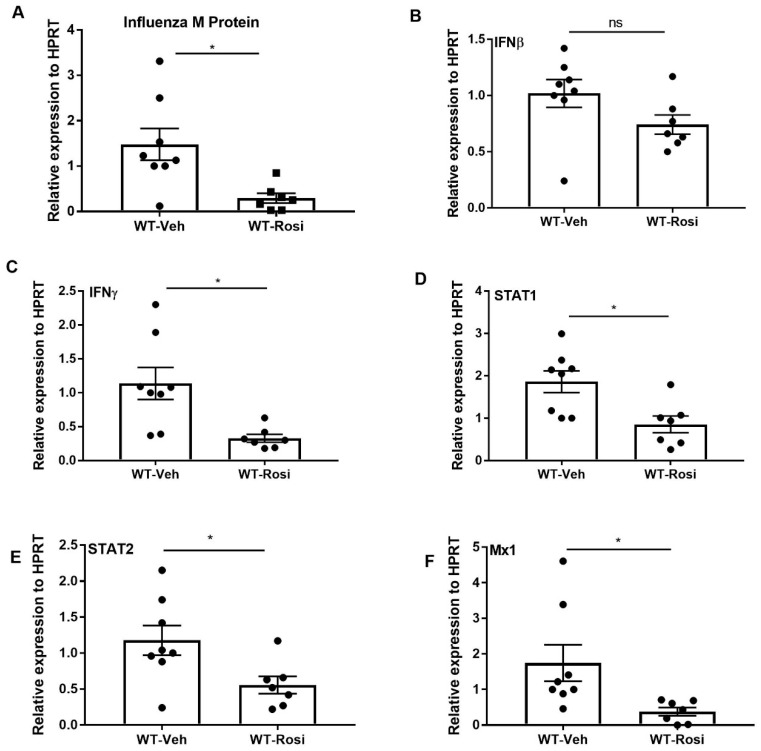
Rosiglitazone treatment decreases influenza viral burden, interferons, and ISG levels during influenza infection. WT male, 6–8 weeks old mice were infected with 100 pfu of influenza (*N* = 7–8 per group). Mice were treated with rosiglitazone or vehicle (DMSO) from day 0–6 post-infection and harvested on day 7 post-infection. (**A**) Viral burden was measured by influenza M protein expression in lung by RT PCR, *N* = 7–8 per group. (**B**–**F**) IFNβ, IFNγ, STAT1, STAT2, and Mx1 expression was measured in whole lung by RT-PCR, *N* = 7–8 per group. Data are represented as mean ±SEM, two tailed Student’s *t* test, * *p* < 0.05, ns-not significant.

**Figure 4 viruses-11-00505-f004:**
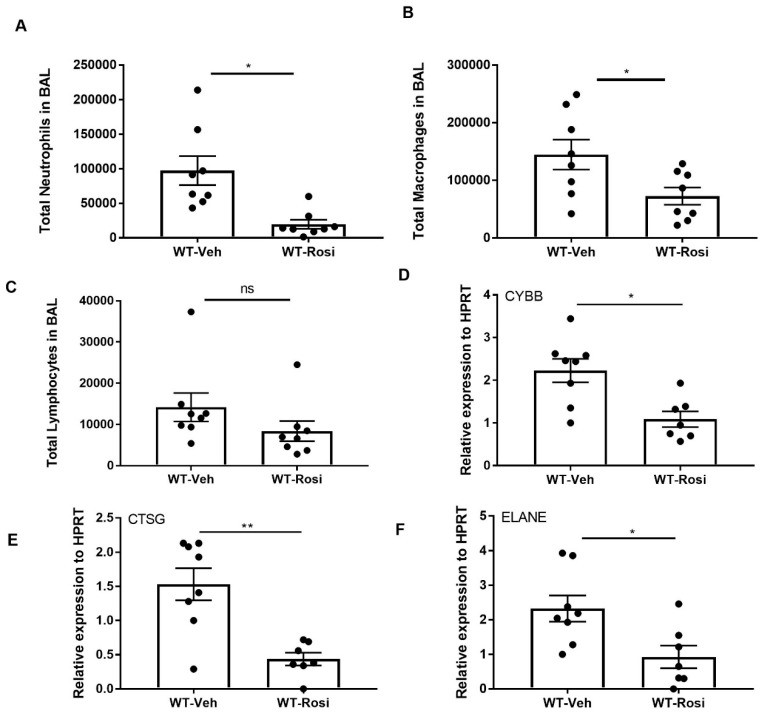
Rosiglitazone treatment decreases the cellular response and genes associated with neutrophil activity during influenza infection. WT male, 6–8 weeks old mice were infected with 100 pfu of influenza (*N* = 7–8 per group). Mice were treated with rosiglitazone or vehicle (DMSO) from day 0–6 post-infection and harvested on day 7 post-infection. The BAL cells were stained, and differential cells were counted as described in methods. (**A**) Total number of neutrophils, (**B**) macrophages, (**C**) and lymphocytes were measured in BAL. (**D**–**F**) CYBB, CTSG and ELANE mRNA expression levels were determined by RT-PCR, *N* = 7–8 per group, Data are represented as mean ± SEM, * *p* < 0.05.

**Figure 5 viruses-11-00505-f005:**
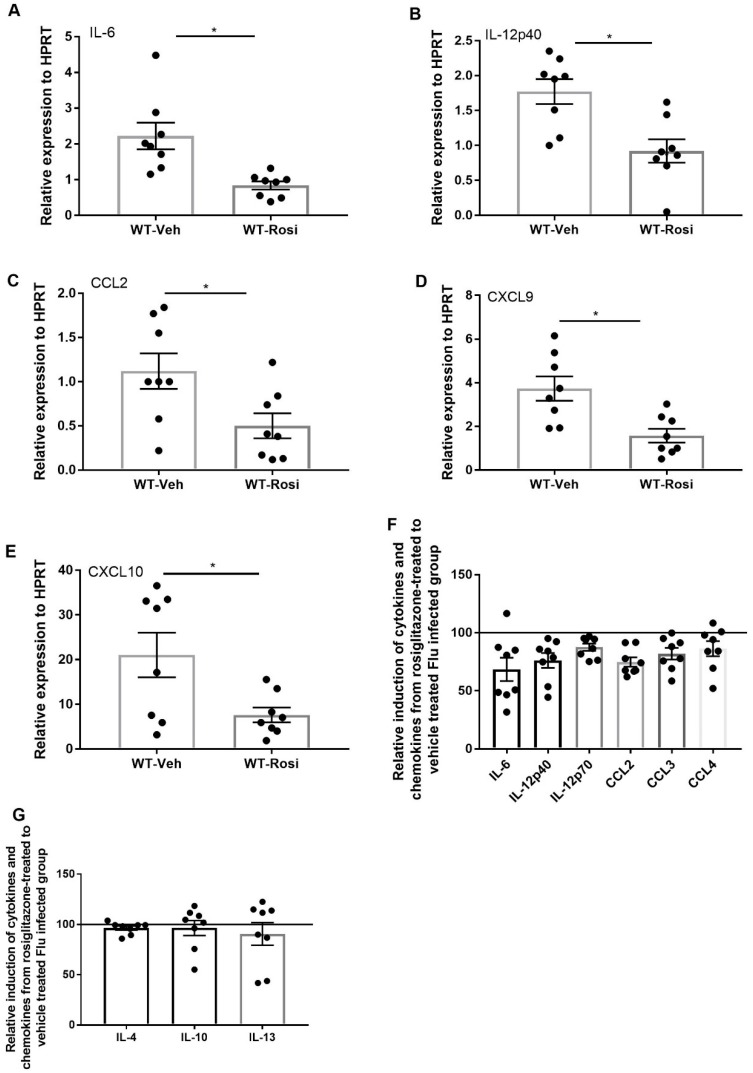
Rosiglitazone treatment decreases the level of inflammatory cytokines and chemokines during influenza infection. WT male, 6–8 weeks old mice were infected with 100 pfu of influenza (*N* = 7–8 per group). Mice were treated with rosiglitazone or vehicle (DMSO) from day 0–6 post-infection and harvested on day 7 post-infection. (**A**–**E**) IL-6, IL-12p40, CCL2, CXCL9 and CXCL10 expression levels were measured by RT-PCR, *N* = 7–8 per group. Relative induction of (**F**) IL-6, IL-12p40, IL-12p70, CCL2, CCL3, and CCL4 (**G**) IL-4, IL-10, and IL-13 levels in response to rosiglitazone-treated to vehicle-treated group by Luminex assay *N* = 7–8 per group, Data are represented as mean ± SEM, * *p* < 0.05, ns-not significant.

**Figure 6 viruses-11-00505-f006:**
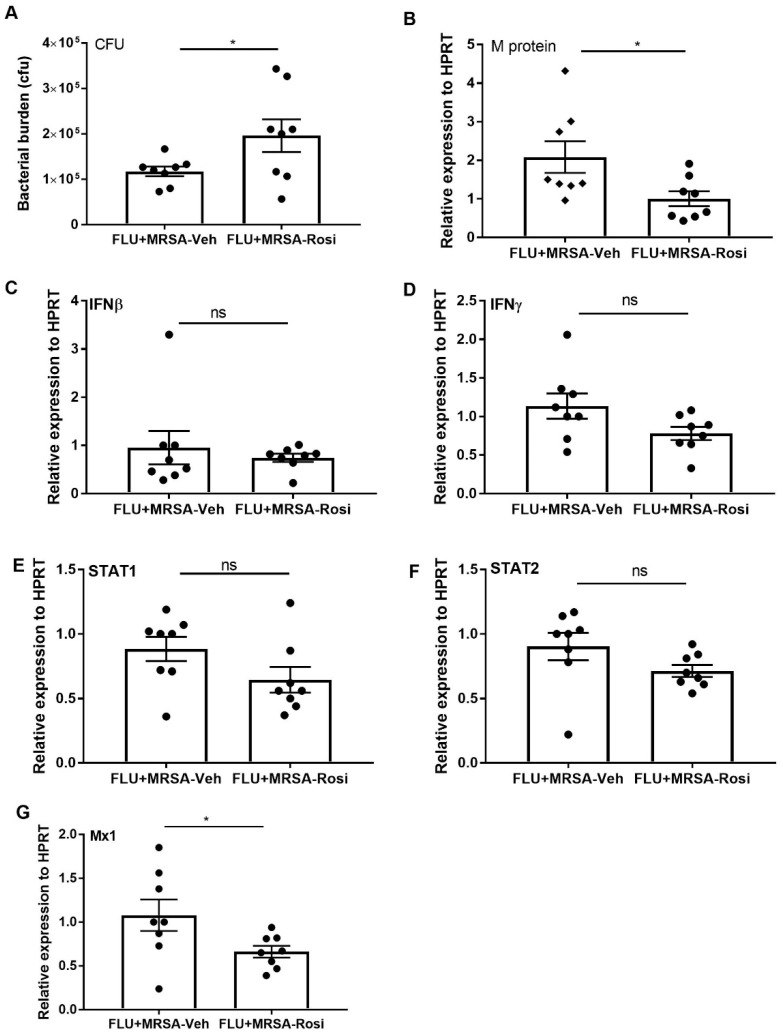
Rosiglitazone treatment decreases bacterial clearance during influenza-MRSA super-infection. WT mice were infected with influenza or PBS for 6 days then challenged with MRSA for one additional day and harvested one day after MRSA infection. Mice were treated with rosiglitazone or vehicle (DMSO) from day 0–6 of influenza infection or PBS treatment. (**A**) Bacterial burden was measured from lung homogenate samples by bacterial plating. (**B**) Viral burden was measured by influenza M protein expression in lung by RT PCR, *N* = 7–8 per group. (**C**–**G**) IFNβ, IFNγ, STAT1, STAT2, and Mx1 expression levels were measured by RT-PCR, *N* = 7–8 per group. Data represented as mean ± SEM, * *p* < 0.05, ns-not significant.

**Figure 7 viruses-11-00505-f007:**
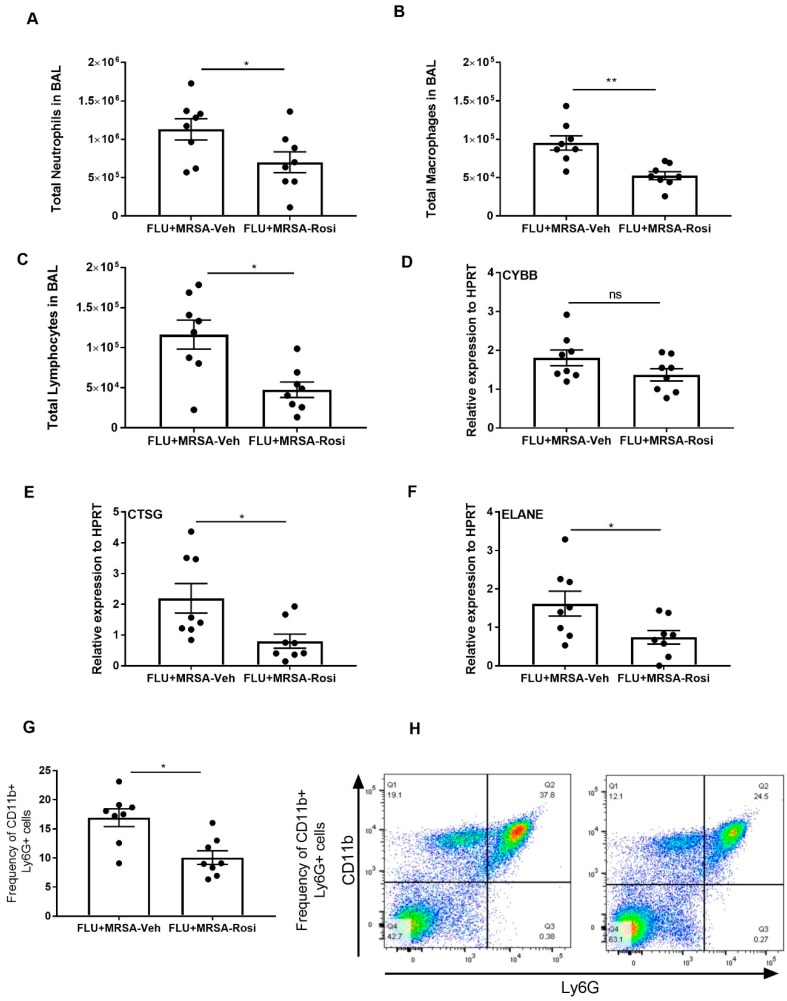
Rosiglitazone treatment decreases the cellular response and genes associated with neutrophil activity during influenza-bacterial super-infection. WT mice were infected with influenza or PBS for 6 days then challenged with MRSA for one additional day. Mice were treated with rosiglitazone or vehicle (DMSO) from day 0-6 of influenza infection or PBS treatment and harvested one day after MRSA infection. The BAL cells were stained, and differential cells were counted as described in methods. (**A**) Total number of neutrophils, (**B**) macrophages, (**C**) and, lymphocytes were measured in BAL. (**D**–**F**) CYBB, CTSG and ELANE mRNA expression levels from lungs were determined by RT-PCR. (**G**) Frequency of CD11b^+^ from BAL cells were determined from influenza and MRSA super infection by flow cytometry (**H**) The representative figures were shown. *N* = 7–8 per group, Data are represented as mean ±SEM, * *p* < 0.05, ** *p* < 0.001, ns-not significant.

**Figure 8 viruses-11-00505-f008:**
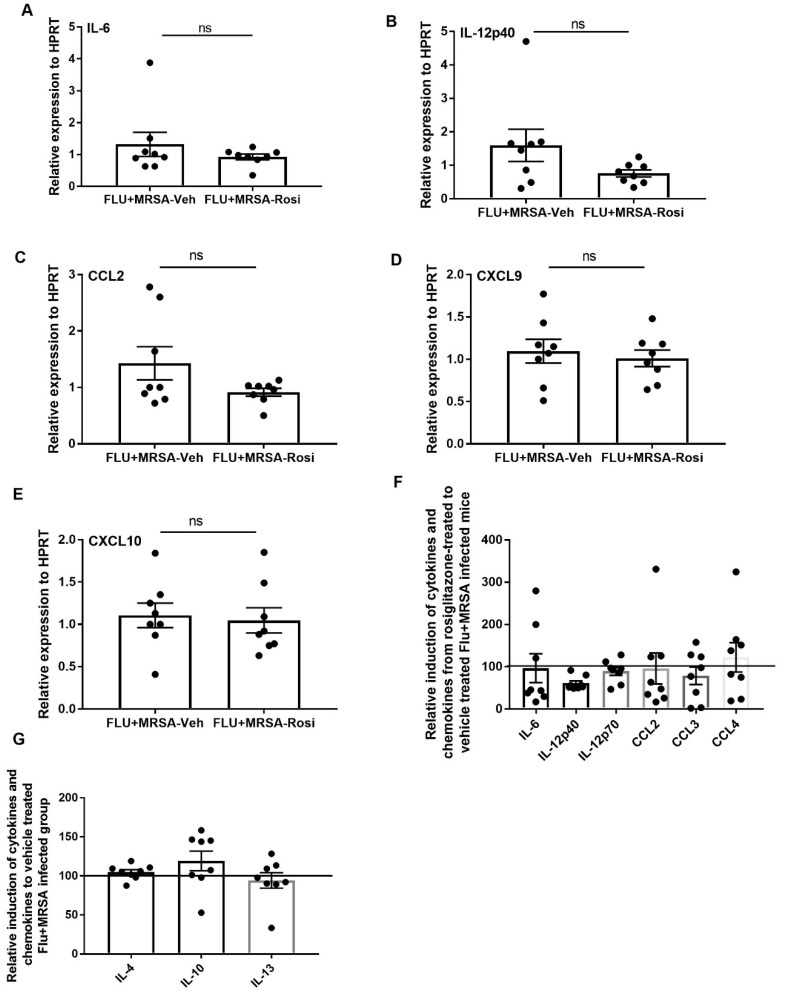
Rosiglitazone treatment does not affect production of inflammatory cytokines and chemokines during influenza-bacterial super-infection. WT mice were infected with influenza or PBS for 6 days then challenged with MRSA for one additional day, and harvested one day after MRSA infection. Mice were treated with rosiglitazone or vehicle (DMSO) from day 0–6 of influenza infection or PBS treatment, (*N* = 7–8 per group). (**A**–**E**) IL-6, IL-12p40, CCL2, CXCL9 and CXCL10 expression levels were measured by RT-PCR, *N* = 7–8 per group. Relative induction of (**F**) IL-6, IL-12p40, IL-12p70, CCL2, CCL3, and CCL4 (**G**) IL-4, IL-10 and IL-13 levels in response to rosiglitazone-treated to vehicle-treated group by Luminex assay *N* = 7–8 per group, Data are represented as mean ± SEM, ns-not significant.

**Table 1 viruses-11-00505-t001:** Baseline characteristics of the study participants (*N* = 6,600).

	All Participants	Rosiglitazone		
Characteristics		No	Yes	P
Prevalence, %	100	95.4	4.6	
Age groups, %				0.27
20–39 years	10.9	11.0	8.9	
40–59 years	38.6	38.3	44.5	
≥ 60 years	50.5	50.7	46.5	
Gender, %				0.80
Men	48.3	48.3	47.4	
Women	51.7	51.7	52.6	
Race/ethnicity, %				0.57
Non-Hispanic Whites	64.0	63.8	67.0	
Non-Hispanic Blacks	16.9	16.9	15.8	
Mexican-Americans	12.3	12.3	12.8	
Other	6.9	7.0	4.4	
Poverty-income ratio, % ^a^				0.08
≤1	16.8	16.9	15.4	
1< & ≤ 3	45.7	46.0	39.1	
>3	37.5	37.1	45.5	
Cigarettes smoking, % ^b^	53.7	54.0	48.6	0.14
Asthma or COPD, % ^c^	17.1	17.0	19.0	0.52
Insulin treatment, %	22.2	21.8	29.8	0.03
Other oral antidiabetic drugs, %	27.3	39.0	65.1	<0.001
Rate of mortality from Influenza and pneumonia (1,000 person-years)	0.8	0.7	2.0	0.02

^a.^ Number of participants with missing data on poverty-income ratio = 730. ^b.^ Number of participants with missing value on smoking = 8. ^c.^ Number of participants with missing data on asthma or COPD = 66. *P*-value for the difference in characteristics calculated using chi-square (*χ^2^*) test.

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
