# Peer review of "Peroxisome Proliferator-Activated Receptor Gamma (PPARγ) Suppresses Inflammation and Bacterial Clearance during Influenza-Bacterial Super-Infection"

_viruses, 2019, doi:10.3390/v11060505_

Round 1
Reviewer 1 Report
Gopal et al. described the PPARγ signaling plays an important role to regulate influenza superinfection and treatment with PPARγ agonist (rosiglitazone) ameliorates inflammatory immune response, influenza virus burden but enhances morality. It is interesting to know the correlation between rosiglitazone treatment and influenza superinfection and also offers information for clinical use of rosiglitazone for treatment. However, many descriptions and findings in the draft were unclear and some important controls were missing.
Major:
1. The title “ Peroxisome Proliferator-activated Receptor gamma (PPARγ) suppresses inflammation and bacterial control during influenza bacterial super-infection” what is the meaning of “bacteria control” ? Similar problems were also shown in line 54-56.
2. In table 1, the result shows that the risk of other oral anti-diabetes drug was more significant than “Rate of mortality from Influenza and pneumonia” in comparison with/without Rosiglitazone treatment. It is not clear why authors just claimed the importance of the rate of mortality from Influenza and pneumonia? The odds ratio and 95%CI were suggested to be analyzed and shown.
3. Some phenomena and results are over-interpreted by the authors. Such as “These data 179 demonstrate that PPARγ expression is decreased during influenza infection and suggest that exogenous PPARγ agonists may have an impact on inflammation (line 178-180). The figure 2 just describes the correlation between PPARγ agonist treatment and influenza infection whereas data of anti-inflammation is lacking.
“The anti-inflammatory response induced by PPARγ agonist during influenza infection is not consistent with increased rosiglitazone associated mortality seen in humans(line 237-238) ” But in table 1, we did not obtain any result regarding anti-inflammatory.
4. The mice challenged with Flu or Flu+MRSA were lacking some important controls. Such as : Mock infected mice treated or untreated with the rosiglitazone as the controls for Figure 3-5 and Mice challenged with Flu or MRSA alone with/without rosiglitazone treatment for Figure 6-7. Without these controls, the data is not so convincing and reliable.
5. In Figure 3A and 6B. If the chemokines and cytokines are reduced, why the viral burden is reduced? Normally, virus should replicate well in absence of immune attack and it is more correlated with high mortality as shown in the Figure 1. Does rosiglitazone treatment directly or indirectly affect Flu replication?
6. Authors’ results indicate treatment of rosiglitazone reduced Flu burden that was found in Flu challenge mice model but bacterial burden was still observed in super-infected mice model. It is unclear that when Flu replication is reduced and airway inflammation turning down, is this environment still beneficial for bacteria invasion and cause secondary infection?
7. In figure 8, results from Flu superinfection mice model indicated that no significance was found in pro-inflammatory cytokine and chemokines. Is it possible that bacterial infection may trigger inflammation that is being reduced by treatment of PPARγ agonist?
8. In mice challenged with Flu or Flu+MRSA, the mice survival rate and body weight changes should be shown.
Minor:
1. Keywords listed that MRSA and Chemokine but these words were not found the abstract.
2. Line 194, interferon stimulated genes(Mx1) should revise to interferon stimulated genes(ISG) including Mx1 or such as Mx1.
3. For the mice study, it is not clear that why the author chose MRSA instead of S. aureus ?
4. All the figures did not mention that which day the challenge mice were collected to be analyzed ?
5. Why the authors select cytochrome B (CYBB), cathepsin G (CTSG) and neutrophil elastase (ELANE) for evaluation of neutrophil activity should be mentioned.
6. Line 219, N=7=8 , please revise to N=7-8.
7. The rationale of analytic candidates of cytokines or chemokines that authors selected should be addressed.
8. Fig 7H, the arrow head is wrong direction
Reviewer 2 Report
Manuscript ”Peroxisome Proliferator-activated Receptor gamma (PPARγ) suppresses inflammation and bacterial control during influenza bacterial super-infection” address the viral infection followed by the bacterial infection and the role of rosiglitazone, an anti-diabetic drug very popularly used among diabetic patients. The authors suggests that the rosiglitazone treatment worsens the outcome of influenza associated pneumonia. PPAR gamma, as a transcription factor has different functions including the modulation of lipid and glucose metabolism, cellular differentiation and inflammation. This manuscript is well written, and has well designed experiments. The data presented are well organized.
My comments and suggestions listed below.
Figure 2, PPAR g levels are compared for the different days of infection, but to correlate this data to the influenza infection efficiency, authors should show the viral (pfu) for each time point. Viral data will substantiate the PPARg level to pfu, and their correlation to the rest of the study.
Figure 3, did authors notice any change in body weight during the administration of rosiglitazone?
Authors need to show the protein levels (data presented are mostly gene expression) that are important in functional analysis, using Elisa or western blot or Luminex multiplexing.
Can we expect a similar result if bacterial infection is followed by viral infection with rosiglitazone treatment?
Line 345- that is repeated –delete it.
Very relevant reference may be added: Cell Host Microbe. 2018 Aug 8;24(2):261-270.e4. doi: 10.1016/j.chom.2018.07.001. Epub 2018 Jul 26. Peroxisome Proliferator-Activated Receptor γ Is Essential for the Resolution of Staphylococcus aureus Skin Infections.
Round 2
Reviewer 1 Report
Most questions are addressed properly. There are still some minor questions that need further revised.
1. The authors indicate influenza infection caused down-regulation of PPARγ. However, this correlation just can be found on day 4 postinfection (figure 2). The virus titers on day 8 and day 12 were almost cleared but PPARγ still kept in significant lower status compared to the mock. The author should explain the possible reasons that led to this phenomenon.
2. Legends in Figure 2 should be revised, “N=5=8 per group” to “N=5-8 per group”.
3. What is Mx1 ? It should be described in the main text.
4. Line 379, the expression of IFNλ ? It should be INF-γ.
Author Response
file attached
